# Protocol for an umbrella review of systematic reviews evaluating the efficacy of digital health solutions in supporting adult cancer survivorship care

Danielle Keane[1*‡], Jean-Paul Calbimonte[2,3‡], Ewa Pawłowska[4,3], Angelos P. Kassianos[5], Joan C. Medina[6], João Gregório[7], Maria Serra-Blasco[8,9], Aleksandar Celebic[10], Antonio Di Meglio[11], Babak Asadi-Azarbaijani[12], Claire Foster[13], Claire L. Donohoe[14], Allini Mafra[15,16], Claudine Backes[15,16], Cristian Ochoa-Arnedo[9,17,18], Derya Gezer[19], Gamze Bozkul[19], Emel Taşvuran Horata[20], Esra Özkan[21], Gillian Prue[22], Gökçe İşcan[23], Gül Dural[24], Gülcan Bahçecioğlu[24], Filiz Ersöğütçü[24], Guna Bērziņa[25], Hicran Bektas[26], Ines- Vaz-Luis[10,27], Izidor Mlakar[28], João Rocha-Gomes[29], Mairead O'Connor[30], Maria Inês Clara[31], Maria Karekla[32], Marte Hoff Hagen[33], Merve Saniye İmançer[34], Oğulcan Çöme[34], Vildan Mevsim[34], Nilay Aksoy[35], Rui Miguel Martins[36], Sıdıka Ece Yokuş[37], Sule Biyik Bayram[38], Aysun Akçakaya Can[38], Tânia Brandão[39], Mohamad M. Saab[1], Nuray Bayar Muluk[40], Zeynep Yıldırım[41], Ioana R. Podina[42], Songül Karadağ[43], Sevilay Erden[43], Remziye Semerci[44], Aydanur Aydin[45], Maximos Frountzas[46], Şengül Üzen Cura[47], Aydın Ruveyde[48], Antonios Billis[49], Jean Calleja-Agius[50], Katarina Vojvodic[51], Poonam Jaswal[1], Eda Sahin[52], Ayşegül Ilgaz[53], Sophie Pilleron[54‡], Josephine Hegarty[1,55‡]

1 Catherine McAuley School of Nursing and Midwifery, University College Cork, Cork, IRELAND, 2 University of Applied Sciences and Arts Western Switzerland HES-SO, Sierre, Switzerland, 3 The Sense Innovation & Research Center, Lausanne, Switzerland, 4 Department of Oncology and Radiotherapy, Faculty of Medicine, Medical University of Gdansk, Gdansk, Poland, 5 Department of Nursing, Cyprus University of Technology, Limassol, Cyprus, 6 Department of Psychology and Education Sciences, Universitat Oberta de Catalunya, Barcelona, Spain, 7 CBIOS – Universidade Lusófona's Research Center for Biosciences & Health Technologies, Lisboa, Portugal, 8 Psycho-oncology and Digital Health Group, Health Services Research in Cancer, Institut d'Investigació Biomèdica de Bellvitge (IDIBELL), L'Hospitalet del Llobregat, Barcelona, Spain, 9 Psycho-Oncology and Digital Health Group, Institut d'Investigació Biomèdica de Bellvitge (IDIBELL), Barcelona, Spain, 10 Institute of Oncology, Clinical Center of Montenegro, Medical School of University of Montenegro, Podgorica, Montenegro, 11 Cancer survivorship program - Inserm Unit 981: Molecular Predictors and New Targets in Oncology, Gustave Roussy, Villejuif, France, 12 Faculty of Health Sciences, VID Specialized University, Oslo, Norway, 13 Centre for Psychosocial Research in Cancer: CentRIC, Health Sciences University of Southampton, Southampton, UK, 14 Department of surgery, Trinity St James Cancer Institute, St James Hospital, Dublin Republic of Ireland, 15 Department of Precision Health, Registre National du Cancer, Luxembourg Institute of Health (LIH), Strassen, Luxembourg, 16 Department of Precision Health, Cancer Epidemiology and Prevention Group (EPI CAN), Luxembourg Institute of Health (LIH), Strassen, Luxembourg, 17 eHealth ICOnnecta't Program and Psycho-Oncology Service, Institut Català d'Oncologia, Barcelona, Spain, 18 Department of Clinical Psychology and Psychobiology, Universitat de Barcelona, Barcelona, Spain, 19 Faculty of Health Science, Nursing Department Tarsus University, Mersin, Türkiye, 20 Department of Physiotherapy and Rehabilitation, Faculty of Health Science, Sakarya University of Applied Science, Sakarya, Türkiye, 21 Nursing Department, Faculty of Health Sciences, Giresun University, Giresun University, Surgical Diseases Nursing. Piraziz, Giresun, Türkiye, 22 School of Nursing and Midwifery, Medical Biology Centre, Queen's University Belfast, Belfast, 23 Suleyman Demirel University, Department of Family Medicine Isparta, Türkiye, 24 Department of Nursing, Faculty of Health Science, Fırat University, Elazığ, Türkiye, 25 Department of Rehabilitation, Riga Stradiņš University, Riga, Latvia, 26 Department of Internal Medicine Nursing, Akdeniz University Faculty of Nursing, Antalya, Türkiye, 27 Interdisciplinary Department for the Organization of Patient Pathways (DIOPP), Gustave Roussy, Villejuif, France, 28 Faculty of Electrical Engineering and Computer Science, University of Maribor, Maribor, Slovenia, 29 Department of Community Medicine, Health Information and Decision, Faculty of Medicine, University of Porto, Porto, Portugal, 30 The National Screening Service, Dublin,

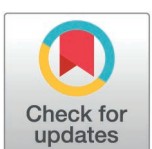

**Data availability statement:** No datasets were generated or analyzed during the current study. All relevant data from this study will be made available upon study completion.

**Funding:** This work has been completed as part of a European Cooperation in Science and Technology (COST Association) project titled: CA21152 Implementation Network Europe for Cancer Survivorship Care (INE-CSC). The views expressed in this paper are of the authors of the paper and not the EU Commission.

**Competing interests:** have read the journal's policy and the authors of this manuscript have the following competing interests:This work has been completed as part of a European Cooperation in Science and Technology (COST Association) project titled: CA21152 Implementation Network Europe for Cancer Survivorship Care (INE-CSC). The views expressed in this paper are of the authors of the paper and not the EU Commission.

Ireland, **31** Faculty of Psychology and Educational Sciences, University of Coimbra, Coimbra, Portugal, **32** Department of Psychology, University of Cyprus, Nicosia, Cyprus, **33** Department of Computer Science, Norwegian University of Science and Technology, Trondheim, Norway, **34** Department of Family Medicine, Faculty of Medicine, Dokuz Eylul University, Izmir, Türkiye, **35** Department of Clinical Pharmacy, School of Pharmacy, Altinbas University, Istanbul, Türkiye, **36** Department of Surgery, Instituto Português de Oncologia de Coimbra, E.P.E., Coimbra, Portugal, **37** Department of Family Medicine, Faculty of Medicine, Manisa Celal Bayar University, Manisa, Türkiye, **38** Karadeniz Technical University, Faculty of Health Sciences, Nursing Department, Trabzon, Türkiye, **39** William James Center for Research, Ispa - Instituto Universitário Lisbon, Portugal, **40** Department of Otorhinolaryngology, Faculty of Medicine, Kırıkkale University, Kırıkkale, Türkiye, **41** Department of Nursing, Faculty of Health Sciences, Ardahan University, Ardahan, Türkiye, **42** Faculty of Psychology and Educational Sciences, Laboratory of Cognitive Clinical Sciences, University of Bucharest, Bucharest, Romania, **43** Department of Nursing Adana/Türkiye, Faculty of Health Sciences, Cukurova University, Adana, Türkiye, **44** Department of Pediatric Nursing, School of Nursing, Koç University, Istanbul, Türkiye, **45** Department of Surgical Nursing, Faculty of Health Sciences, Gumushane University, Gumushane, Türkiye, **46** Medical School, National and Kapodistrian University of Athens, Greece, **47** Nursing Department, Faculty of Health Science, Çanakkale Onsekiz Mart University, Çanakkale, Türkiye, **48** Nursing Department, Faculty of Health Sciences, Ondokuz Mayıs University, Samsun, Türkiye, **49** Lab of Medical Physics & Digital Innovation, School of Medicine, Aristotle University of Thessaloniki, Thessaloniki, Greece, **50** Department of Anatomy, Faculty of Medicine and Surgery, University of Malta, Msida, Malta, **51** Institute of Public Health of Belgrade, Belgrade, Serbia, **52** Faculty of Health Sciences, Giresun University, Giresun, Türkiye, **53** Department of Public Health, Nursing, Akdeniz University, Antalya, Türkiye, **54** Department of Precision Health, Ageing, Cancer, and Disparities Research Unit, Luxembourg Institute of Health, Strassen, Luxembourg, **55** Cancer Research @UCC, College of Medicine & Health, University College Cork, Cork, Ireland

‡ DK and JPC are Joint first authors. SP and JH are Joint last author.
* daniellekeane@ucc.ie

## Abstract

### Introduction

The growing number of people living with, through and beyond cancer poses a new challenge for sustainable survivorship care solutions. Digital health solutions which incorporate various information and communication technologies are reshaping healthcare; offering huge potential to facilitate health promotion, support healthcare efficiencies, improve access to healthcare and positively impact health outcomes. Digital health solutions include websites and mobile applications, health information technologies, telehealth solutions, wearable devices, AI-supported chatbots and other technologically assisted provision of health information, communication and services. The breadth and scope of digital health solutions necessitate a synthesis of evidence on their use in supportive care in cancer. This umbrella review will identify, synthesise, and compare systematic reviews which have evaluated the efficacy or effectiveness of digital solutions for adult cancer survivorship care with a particular focus on surveillance and management of physical effects, psychosocial effects, new cancer/recurring cancers and supporting health promotion and disease prevention.

### Methods and analysis

An umbrella review of published systematic reviews will be undertaken to explore the types of digital health solutions used, their efficacy or effectiveness as a form of

supportive care, and the barriers and enablers associated with their implementation. The umbrella review will be reported according to the Preferred Reporting Items for Overviews of Reviews (PRIOR) checklist. A search will be conducted across key databases. Records will be assessed independently by two review authors for eligibility against predefined criteria and will undergo two stage title, abstract and full text screening. All systematic reviews that meet the inclusion criteria will be assessed for quality using the AMSTAR 2 checklist with quality assessment and data extraction by two reviewers. The degree of publication overlap of primary studies across the included reviews will also be calculated and a mapping of the evidence will also be presented.

## Ethics and dissemination

As this research proposes using systematic reviews that are already published, ethical approval is not required. Results from this umbrella review will be published in a peer-reviewed journal where any significant deviations from the protocol will be justified.

## Introduction

The number of individuals living with, through and beyond cancer is growing each year, largely due to considerable advancements in prevention, early detection, diagnosis, treatment and follow-up [1]. As a result, cancer is increasingly recognized as a chronic and ongoing condition, requiring long-term management. Cancer survivors represent a large and heterogeneous group, encompassing individuals from the moment of diagnosis throughout the balance of their life [2].

Cancer and its treatment can significantly affect an individual's quality of life; with impacts varying substantially by the health, disease and socio-economic status of cancer survivors [3–5]. The physical effects of the disease and its treatments, combined with the emotional and psychological challenges can lead to a range of difficulties including fatigue, pain, emotional distress, and changes in social relationships [6–9]. These ongoing effects highlight the need for comprehensive support to improve the functional capacity and well-being of individuals living with and beyond cancer.

Digitally assisted self-management support and targeted digital interventions have demonstrated positive effects on patient outcomes, particularly when they are part of the cancer survivorship care pathway [10–14]. While there are advantages to healthcare professional led interventions, such as increased adherence to the intervention, many healthcare professionals are already time-constrained and may not have training in empirically supported behavioural and counselling skills. Moreover, inequalities exist in the distribution of healthcare professionals globally, leading to underserved populations or regions [15]. Accessing supportive care can be challenging for many reasons including the fatigue and symptom burden associated with cancer; challenges with transportation and language; time and financial constraints; health literacy issues; and in some instances, patients being too embarrassed to ask for help from their health care professional [16–19]. These factors have prompted increased interest in understanding the barriers and enablers to the implementation of digital solutions [20,21].

According to the European Commission digital health and care refers "*to tools and services that use information and communication technologies to improve prevention, diagnosis, treatment, monitoring and management of health-related issues and to monitor and manage lifestyle-habits that impact health [...] improve access to care and the quality of that care [...] and increase the overall efficiency of the health sector*" [22]. These can include mobile health (mHealth), e-health, health information technology, telehealth, telemedicine, platforms, apps, wearable devices, sensors, chatbots and the use of digital solutions.

The benefits of digital health solutions in a cancer context include providing mechanisms of integrating care across primary, secondary and tertiary healthcare systems and supporting the provision of a person-centred and personalised care approach [23,24]. Digital solutions for survivorship support are attractive because they have the potential to be provided

to many cancer survivors simultaneously using less resources, potentially resulting in lower associated carbon emissions than in-person care [25–27]. Virtual or remote care offers individuals the opportunity to access their care at their preferred time and place, removing constraints associated with the burden of illness and leading to increased engagement in behaviours that enhance their quality of life [28,29]. Recent studies have also shown that digital interventions can improve self-management behaviours and increase confidence to self-manage, including adherence to after-treatment care and medication [30–32]. Some digital health solutions can also have a positive effect on overall survival of cancer patients [33,34].

However, the widespread reach, adoption, and implementation of digital health solutions may be hampered by inadequate digital health systems infrastructure; heterogeneity in digital health solutions and healthcare systems; unequal access to digital solutions; issues with internet connections; limits in digital literacy and the lack of synthesized evidence on the effectiveness of digital interventions in a healthcare context [21,35,36]. Some enablers have also been emphasized in the literature including co-design of digital solutions; being aware of users' literacy level; use of interventions and tools that respect the cultural attributes and the risk of social health inequalities of future service users [21,37,38].

The lack of a standardised methodology to evaluate the plethora of available digital solutions for survivorship support in real-world settings further complicates evaluation of efficacy of digital solutions [39–41]. To ensure quality care in cancer survivorship, it is critical to distinguish between digital solutions in a health care context which have or have not undergone rigorous testing to ensure safety and efficacy; meet legislated privacy and security requirements and offer clinically meaningful benefits [42]. Healthcare professionals, patients, and users need to be supported to make more informed decisions about selecting and utilising high-quality and reliable digital solutions.

Given the rapid advancement of digital solutions, a systematic evaluation of the existing evidence is essential to evaluate the effectiveness of digital solutions in supporting cancer survivorship care. Additionally, identifying key enablers and barriers to their implementation will be crucial for optimizing their integration into healthcare systems.

## Aim

This umbrella review aims to identify, synthesise, and compare evidence from published systematic reviews that have evaluated the efficacy or effectiveness of digital solutions designed to support cancer survivorship care in adults.

### Review questions

This umbrella review seeks to address the following questions:

Q1.  What patient-oriented digital health solutions for supporting cancer survivorship care for adults have been tested within the included systematic reviews?

Q2.  How do the outcomes presented in each systematic review map onto Nekhlyudov et al., (2019) Quality of Cancer Survivorship Care Framework domains [43]?

Q3a.  What impact do the digital health solutions for supporting cancer survivorship care have on the primary outcomes associated with the domains and indicators described in the Quality of Cancer Survivorship care Framework [43] (i.e., prevention and surveillance of new cancer/ recurring cancers; surveillance and management of physical effects; surveillance and management of psychosocial effects)?

Q3b.  What impact do the digital health solutions for supporting cancer survivorship care have on the secondary outcomes associated with the domains and indicators described in Quality of Cancer Survivorship Care Framework [43] (i.e., surveillance and management of chronic medical conditions; health promotion and disease prevention)?

Q4.  What is the degree of overlap of included primary studies across the included systematic reviews? Are there areas of homogeneity or heterogeneity in the evidence and potential biases?

Q5. What are the key considerations, barriers, and enablers to the implementation of patient-oriented digital solutions to support adult cancer survivorship care articulated by the authors of the included systematic reviews?

Q6. What are the gaps and limitations of the evidence-base highlighted by review authors?

## Materials and methods

### Design

An umbrella review methodology will be utilised for this research as it allows for the integration, examination, and comparison of evidence from existing systematic reviews in a concise format [44,45]. As the area of digital health technology is broad and varied, an umbrella review is a particularly suitable methodology, as it can impose coherence on evidence by separating the topic into targeted populations, interventions, or both [46]. The umbrella review will be carried out in accordance with the Preferred Reporting Items for Overviews of Reviews (PRIOR) checklist (S1 Checklist) [47].

### Eligibility criteria

Detailed eligibility criteria using the population, intervention, context, and outcome (PICO) framework [48] are outlined within this section and in tabular format within S1 Table.

**Population.** Individuals who have received a diagnosis of cancer and are defined by the author as adults, usually 18 years and above.

**Intervention.** The intervention of interest is the use of patient-oriented digital health solutions integrated into the cancer survivorship supportive care journey, with a focus on supporting patients in achieving the outcomes identified in this review.

Due to their novel nature and breadth of utility, there are many varying definitions and descriptions of what encompasses a digital health solution, some of which are presented in S2 Table. Despite variation, definitions of digital health solutions contain overlapping elements and typically involve a focus on the use of information and communication technologies or tools or solutions to facilitate improved or more streamlined access to care; enhanced quality or greater efficiency in health care delivery processes through enhanced data collection, data sharing and/or communication between the patient and healthcare professional; and direct to patient health promoting advice and interventions, self-management support and therapeutics[22].

**Comparison.** Comparisons will be presented as per the included publications. Studies with and without a comparator will be considered for inclusion.

**Outcome.** Outcomes will be classified according to the domains of the Quality of Cancer Survivorship Care Framework [43], including the associated outcomes for each domain (Table 1 and 2). Details of these are provided in an expanded format in (S3 TableA and B).

**Publication type and design.** This umbrella review will take into consideration systematic reviews from any country or region that incorporate experimental studies [randomised controlled trials or non-randomised studies) where the clear primary aim is to assess the efficacy or effectiveness of the digital health solutions in terms of named outcomes. Systematic reviews having a mixture of, or only incorporating studies with feasibility, acceptability, satisfaction, pilot study aims will be excluded.

### Information sources and search strategy

The search will be conducted in CINAHL Plus with full text, ERIC, MEDLINE, APA PsycArticles, APA PsycInfo, SocINDEX with full text via EBSCOhost Research Databases interface; Web of Science; Embase, IEEE Xplore and the Cochrane library. Keywords and search strings will be used as follows: cancer, survivorship, digital, and systematic review. These will be combined using Boolean operators and relevant proximity indicators. Subject headings will be used as appropriate. The detailed search strategy for a sample database can be found in S4 Table.

**Table 1. Primary outcome domains adapted from the Quality of Cancer Survivorship Care Framework [43].**

| Domain | Definition of outcome data |
|---|---|
| Prevention and surveillance for recurrence and new cancers | Reports of actions taken to lower risks of developing cancer and/or their impact. Reports of data collected and analysed relating to the monitoring for or the development of recurrence of cancer or a new cancer. |
| Surveillance and management of physical effects | Reports of the assessment of physical symptoms and/or conditions via history, physical examination, and/or standardized or self-reported instruments/ tools/ surveys. Reports of the management of symptoms and/or conditions based upon such assessments and/or their impact on physical effects. |
| Surveillance and management of psychosocial effects | Reports of the assessment of psychosocial effects or symptoms and/or conditions via history, and/or standardized instruments/tools. Reports of the management of symptoms and/or conditions based upon such assessments and/or their impact on psychosocial effects. |

**Table 2. Secondary outcome domains, adapted from the Quality of Cancer Survivorship Care Framework [43].**

| Domain | Definition of outcome data |
|---|---|
| Surveillance and management of chronic medical conditions | Reports of assessment of cancer- and noncancer- related symptoms and/or conditions using appropriate screening and treatments. |
| Health promotion and disease prevention | Reports of actions taken to address health determinants such as behaviours and lifestyles and/or their impact. Reports of referral to services that promote evidence-based health behaviours to lower risk of disease and/or their impact. |

## Study records

**Study selection.** Records will be imported into Covidence, an online software used to streamline the production of reviews of the literature [49]. Duplicates will be deleted automatically in Covidence. Two independent authors in pairs, will conduct the screening of the records retrieved by the search strategy for inclusion or exclusion against the outlined eligibility criteria. Each record will first undergo title and abstract screening. Records that potentially meet the inclusion criteria will be retrieved in full-text format. Researchers will review the full texts independently and any screening conflicts will be resolved through discussion with a third author. Authors will provide the list of studies excluded at full text stage and justify the exclusions as a supplementary file.

Screening of non-English language publications will be done by researchers who can speak the language, where possible. If this is not possible, Google translate function will be used and the use of same will be noted where relevant. Final review for eligibility will be based upon the translated abstract.

The quality of the included systematic reviews will be assessed using the A MeaSurement Tool to Assess systematic Reviews 2 (AMSTAR 2) checklist [50]. Two reviewers will independently assess the methodological quality of each systematic review against the AMSTAR 2 rating scale. Using the AMSTAR 2 quality appraisal tool, the quality of reviews will be rated as high, moderate, low, or critically low based on the cumulative rating of 16 domains, of which, seven are considered critical. This includes a registered protocol prior to the review, adequate search of literature, reasons for exclusion, included studies assessed for risk of bias, appropriate methods of meta-analysis, assessing risk of publication bias and bias in result interpretations [50]. All quality assessments will be discussed by the two reviewers to resolve any discrepancies, and a third reviewer will be consulted if necessary.

A widely recognised issue faced when conducting umbrella reviews is the challenge of primary study overlap within included systematic reviews[51]. To calculate the degree of overlap, first, a citation matrix of included systematic reviews and their primary studies will be created. Secondly, we will calculate the corrected cover area (CCA) [52]. The CCA

expressed as a percentage indicates the proportion of overlapping primary studies and is calculated as $(N-r)/((r \times c)-r)$, where N is the total number of primary study occurrences in the citation matrix, r is the number of index primary studies (i.e., number of rows within the citation matrix) and c is the total number of reviews (columns in citation matrix). The CCA can have a value between 0% and 100%, with the value being interpreted as slight (0%–5%), moderate (6%–10%), high (11%–15%), or very high (>15%) [50]. We will consider the limitations of the approach [52] and not exclude any review based upon this tool but simply highlight the overlap graphically and numerically.

## Data collection, management, and organisation

A PRISMA flow diagram will be used to represent study identification and selection [53]. Selected studies will be saved into a bibliography management software (e.g., Zotero) [54] for citation tracking.

Several researchers/ co-authors will be involved in data extraction, working in pairs. The process will be carried out according to the Joanna Briggs Institute guidelines for the conduct of umbrella reviews [55]. Data items will include characteristics of systematic review and included papers as follows: author and year; synthesised participant details (cancer types, gender/sex, mean/median age if available at review level); type of review (systematic review; systematic review and meta-analysis, systematic review and meta-synthesis; systematic review and meta-summary; systematic review and network analysis, other); numbers of databases searched; search date range; numbers and design(s) of included studies; countries (n) and continents represented by included studies; appraisal instrument(s) used; and method of synthesis/ meta-analysis. Details pertaining to any meta-analysis (direct or network meta-analyses or other), details of outcomes and parameters used for pooling, publication bias assessment, type of outcome metrics (Risk Ratio or Odds Ratio), meta-analysis model, the summary meta-analytic estimate and corresponding 95% confidence interval (CI), and heterogeneity measure will also be extracted. For systematic reviews without meta-analysis, we will describe the number of studies that found statistically significant associations and the direction of the association estimates.

Specific data items relating to the focus of this review will be extracted as follows: the aim pertaining to digital solutions represented in each systematic review; name and type of digital health solutions or interventions represented in the review categorised using an adaptation of Lupton's (2014) [56] typology of digital health technologies (S5 Table); primary and secondary outcomes represented within each review; characteristics of digital health solutions represented in the review associated with positive outcome changes, and narrative regarding overall results of systematic reviews. Categories of digital health interventions will be adapted from Lupton's [56] typology of digital health technologies and will include: telemedicine/ telehealth; e-learning, training and information sharing; digital diagnostics and risk-assessment technologies; health informatics such as electronic patient records; digital health promotion using SMS, apps and other digital technologies; digital monitoring/ tracking devices such as apps, wearable technologies, smartphones; dedicated platforms; digital health games designed for fitness, tracking/ monitoring, health promotion and education; and other (S5 Table).)

Systematic review authors perspectives on enablers and barriers pertaining to the implementation of digital solutions will also be extracted.

Details of the sources of funding for this umbrella review (tertiary source) and the included systematic reviews (secondary sources) will be reported. Details of the funding of the individual studies (primary sources) included in each systematic review will not be reported.

## Data synthesis

The data extracted detailing the characteristics of the included systematic reviews will be summarised in tabular and narrative format. Particular attention will be paid to the population of focus within the included systematic reviews. Where a review focuses on clear categories of diagnosis, i.e., primary cancer, advanced or metastatic cancer, or a mixture of these, data synthesis will be informed by these categories.The systematic reviews will be mapped against the domains and outcomes included in the Quality of Cancer Survivorship Care Framework [43] and Lupton's (2014) [56] typology of

digital health technologies. The Quality of Cancer Survivorship Care Framework domains are based on fundamentals of survivorship care arising from the Institute of Medicines seminal report [57], making it a pertinent and relevant tool for mapping evidence relating to survivorship care, including digital solutions.

Networks are graphical depictions of the relationships between variables of interest, known as nodes which are connected by edges or links. Network analysis will be used to visually model the degree of connection between the outcomes of the included reviews and the domains and outcomes outlined by Nekhlyudov and colleagues [43]. The network analysis will be conducted based on the process presented by Kemp et al. [58] to visualise connections and overlaps in the existing research that may have been previously viewed in isolation. Edges connecting the nodes can be positive or negative (depicted graphically using colour), weighted or unweighted (depicted graphically through the thickness of the edge), to reflect the relationship between the nodes. When a network is created, its graphical rendering shows the relational structure of the nodes which can be useful in analysis. For example, the degree of centrality of certain nodes over others within the network indicates the importance of that node relative to the others [59]. To facilitate the Network Analysis, included systematic reviews will be mapped against the domains and outcomes included in the Quality of Cancer Survivorship Care Framework [43] and Lupton's typology of digital health technologies [56] by:

1. Mapping the aim pertaining to digital solutions represented in each review against the pertinent Quality of Cancer Survivorship Care Framework domains and Lupton's typology of digital health technologies.

2. Mapping the primary and secondary outcomes represented in each review against the Quality of Cancer Survivorship Care Framework domains and outcomes.

3. Conducting a network analysis to elucidate the scope of clustering of reviews, and connectivity across domains and outcomes. Like Kemp et al, [58] two bipartite networks will be created with network one being the included systematic review aims mapped onto the domains and network two being the included systematic reviews' outcomes mapped onto the domains and outcomes of the framework.

4. Exploring details of the communities within each network.

Outcome data extracted from the included systematic reviews will be tabulated and grouped by intervention type. For this umbrella review, the unit of analysis is the systematic review, therefore only review level summary data will be synthesised. It is anticipated that there will be significant heterogeneity in terms of the intervention characteristics, intensity, and duration, outcomes type and outcome measures. Thus, it is expected that we will conduct a narrative synthesis to explore relationships between types and features of the interventions, review findings concerning relational outcomes, and systematic review authors' conclusions about the effects of interventions. This synthesis will follow the "Synthesis without meta-analysis (SWiM) in systematic reviews" guideline [60]. To this end, we will (i) group included reviews according to interventions types, outcomes and study designs; (ii) describe the standardised metric and transformation methods used for outcomes addressed; (iii) describe the synthesis methods used; (iv) outline the criteria used to prioritise results for summary and/or synthesis; (v) outline basis for judgments regarding heterogeneity in reported effects; and (iv) outline methods used to assess certainty of the synthesis findings and describe data presentation methods.

The possible sources of heterogeneity, including variation in PICO; variations in context, design and methodological considerations particularly the inclusion of non-randomised designs will be discussed.

The authors will provide a summary of all the outcomes and associated measurement tools that have been specifically analysed in the included systematic reviews and meta-analyses. If a review has correlated such study endpoints/outcomes to cancer survival or quality of life, this will also be reported.

Data pertaining to barriers and enablers will be summarised using the five domains of innovation, outer setting, inner setting, individual and implementation process within the Consolidated Framework for Implementation Research (CFIR) framework [61].

## Supporting Information

**S1 Checklist. PRISMA-P 2015 Checklist.**
(DOCX)

**S1 Table. Detailed eligibility criteria outlined according to PICO framework.**
(DOCX)

**S2 Table. Definitions of digital health solutions.**
(DOCX)

**S3 Table. A.** Expanded outcome domains adapted from the Quality of Cancer Survivorship Care Framework including associated outcomes for each domain. **B.** Expanded Secondary Outcomes, adapted from the Quality of Cancer Survivorship Care Framework including associated outcomes.
(DOCX)

**S4 Table. Detailed CINAHL sample search.**
(DOCX)

**S5 Table. Categories of digital health solutions, adapted from Lupton's (2014) typology of digital health technologies.**
(DOCX)

## Author contributions

**Conceptualization:** Jean-Paul Calbimonte, Ewa Pawłowska, Angelos P. Kassianos, Gillian Prue, Sophie Pilleron, Josephine Hegarty, Poonam Jaswal, Ayşegül Ilgaz.

**Methodology:** Poonam Jaswal.

**Writing – original draft:** Danielle Keane, Jean-Paul Calbimonte, Ewa Pawłowska, Angelos P. Kassianos, Claire Foster, Babak Asadi-Azarbaijani, Gillian Prue, Maria Inês Clara, Sophie Pilleron, Josephine Hegarty, Poonam Jaswal.

**Writing – review & editing:** Danielle Keane, Jean-Paul Calbimonte, Ewa Pawłowska, Angelos P. Kassianos, Joan C. Medina, João Gregório, Maria Serra-Blasco, Aleksandar Celebic, Allini Mafra, Antonio Di Meglio, Claire Foster, Babak Asadi-Azarbaijani, Claire L. Donohoe, Claudine Backes, Cristian Ochoa-Arnedo, Derya Gezer, Emel Taşvuran Horata, Gamze Bozkul, Esra Özkan, Gillian Prue, Gökçe İşcan, Gül Dural, Gülcan Bahçecioğlu, Filiz Ersöğütçü, Guna Bērziņa, Hicran Bektas, Ines- Vaz-Luis, Izidor Mlakar, João Rocha-Gomes, Mairead O'Connor, Maria Inês Clara, Maria Karekla, Marte Hoff Hagen, Merve Saniye İmançer, Oğulcan Çöme, Vildan Mevsim, Nilay Aksoy, Rui Miguel Martins, Sıdıka Ece Yokuş, Sule Biyik Bayram, Aysun Akçakaya Can, Tânia Brandão, Mohamad M. Saab, Nuray Bayar Muluk, Zeynep Yıldırım, Ioana R. Podina, Songül Karadağ, Sevilay Erden, Remziye Semerci, Aydanur Aydin, Maximos Frountzas, Şengül Üzen Cura, Aydın Ruveyde, Antonios Billis, Jean Calleja-Agius, Katarina Vojvodic, Sophie Pilleron, Josephine Hegarty, Poonam Jaswal, Eda Sahin, Ayşegül Ilgaz.

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
