## [Decision Letter · Decision Letter 0]

2 Feb 2025

PONE-D-24-26581Protocol for an umbrella review of systematic reviews evaluating the efficacy of digital health solutions in supporting adult cancer survivorship care.PLOS ONE

Dear Dr. Keane,

Thank you for submitting your manuscript to PLOS ONE. After careful consideration, we feel that it has merit but does not fully meet PLOS ONE’s publication criteria as it currently stands. Therefore, we invite you to submit a revised version of the manuscript that addresses the points raised during the review process.

We look forward to receiving your revised manuscript.

Kind regards,

Ronald Chow

Academic Editor

PLOS ONE

Journal Requirements:

-https://crimsonpublishers.com/tteh/pdf/TTEH.000582.pdf

In your revision ensure you cite all your sources (including your own works), and quote or rephrase any duplicated text outside the methods section. Further consideration is dependent on these concerns being addressed.

5. Please amend your list of authors on the manuscript to ensure that each author is linked to an affiliation. Authors’ affiliations should reflect the institution where the work was done (if authors moved subsequently, you can also list the new affiliation stating “current affiliation:….” as necessary).

6. Please include captions for your Supporting Information files at the end of your manuscript, and update any in-text citations to match accordingly. Please see our Supporting Information guidelines for more information: http://journals.plos.org/plosone/s/supporting-information .

Reviewers' comments:

Reviewer's Responses to Questions

**Comments to the Author**

1. Does the manuscript provide a valid rationale for the proposed study, with clearly identified and justified research questions?

Reviewer #1: Yes

Reviewer #2: Yes

2. Is the protocol technically sound and planned in a manner that will lead to a meaningful outcome and allow testing the stated hypotheses?

Reviewer #1: Yes

Reviewer #2: Yes

3. Is the methodology feasible and described in sufficient detail to allow the work to be replicable?

Reviewer #1: Yes

Reviewer #2: Yes

4. Have the authors described where all data underlying the findings will be made available when the study is complete?

Reviewer #1: Yes

Reviewer #2: Yes

5. Is the manuscript presented in an intelligible fashion and written in standard English?

Reviewer #1: Yes

Reviewer #2: Yes

6. Review Comments to the Author

You may also provide optional suggestions and comments to authors that they might find helpful in planning their study.

Reviewer #1: The authors are to be commended for designing this umbrella review which plans to synthesize the existing data for digital health solutions for cancer survivorship and identify gaps for future research. The methodology is well thought out and involves a comprehensive and robust analysis of systematic reviews.

I have a two minor comments for the authors to consider:

Patients with early cancers have different supportive care needs compared to those with metastatic cancers. While the definition of “cancer survivor” refer to all patients regardless of stage and the domains of the Quality of Cancer Survivorship Care Framework apply to both groups of patients, it is relevant to categorise which interventions are for those with early cancers, and which are those for advanced or metastatic cancers. Please clarify if there is any plan to make this categorization in the analysis. If there is, the authors are suggested to provide definitions of early versus advanced and/or metastatic cancer for consistency in the categorization.

Unlike survival metrics, analysis of the effectiveness of supportive care measures may have significant variability or heterogeneity. The authors are suggested to provide a summary of all the study endpoints that have been analysed in the systematic reviews and meta-analyses, and evaluate whether they correlated with cancer survival or quality of life. This may help future clinical trials on digital health solutions select appropriate and meaningful outcome measures.

Reviewer #2: This proposed umbrella review will synthesize and summarize the existing literature on the efficacy of digital health solutions in supporting adult cancer survivorship care. This is an important and relevant topic that would benefit from a review of literature.

The protocol is well written. The methodology is detailed and utilizing evidence-based tools such as the PRIOR and AMSTAR 2 checklists appropriately.

I have no suggestions.

7. PLOS authors have the option to publish the peer review history of their article (what does this mean? ). If published, this will include your full peer review and any attached files.

**Do you want your identity to be public for this peer review?** For information about this choice, including consent withdrawal, please see our Privacy Policy .

Reviewer #1: No

Reviewer #2: No

---

## [Author Response · Author response to Decision Letter 1]

14 Mar 2025

Comment:

The authors are to be commended for designing this umbrella review which plans to synthesize the existing data for digital health solutions for cancer survivorship and identify gaps for future research.

The methodology is well thought out and involves a comprehensive and robust analysis of systematic reviews.

I have a two minor comments for the authors to consider

Response:

Thank you for taking the time to review the manuscript. The authorship team appreciate your kind appraisal and welcome the opportunity to implement your suggestions to enhance the manuscript for resubmission.

Thank you

Comment:

Patients with early cancers have different supportive care needs compared to those with metastatic cancers.

While the definition of “cancer survivor” refer to all patients regardless of stage and the domains of the Quality of Cancer Survivorship Care Framework apply to both groups of patients, it is relevant to categorise which interventions are for those with early cancers, and which are those for advanced or metastatic cancers.

Please clarify if there is any plan to make this categorization in the analysis. If there is, the authors are suggested to provide definitions of early versus advanced and/or metastatic cancer for consistency in the categorization.

Response:

Patients with early cancers have different supportive care needs compared to those with metastatic cancers.

While the definition of “cancer survivor” refer to all patients regardless of stage and the domains of the Quality of Cancer Survivorship Care Framework apply to both groups of patients, it is relevant to categorise which interventions are for those with early cancers, and which are those for advanced or metastatic cancers.

Please clarify if there is any plan to make this categorization in the analysis. If there is, the authors are suggested to provide definitions of early versus advanced and/or metastatic cancer for consistency in the categorization. Thank you for sharing this observation. We appreciate this perspective.

As per the suggestion of the reviewer we have added text as follows:

Particular attention will be paid to the population of focus within the included systematic reviews. Where a review focuses on clear categories of diagnosis, i.e. primary cancer, advanced or metastatic cancer, or a mixture of these, data synthesis will be informed by these categories.

Comment:

Unlike survival metrics, analysis of the effectiveness of supportive care measures may have significant variability or heterogeneity.

The authors are suggested to provide a summary of all the study endpoints that have been analysed in the systematic reviews and meta-analyses, and evaluate whether they correlated with cancer survival or quality of life. This may help future clinical trials on digital health solutions select appropriate and meaningful outcome measures.

Response:

Unlike survival metrics, analysis of the effectiveness of supportive care measures may have significant variability or heterogeneity.

The authors are suggested to provide a summary of all the study endpoints that have been analysed in the systematic reviews and meta-analyses, and evaluate whether they correlated with cancer survival or quality of life. This may help future clinical trials on digital health solutions select appropriate and meaningful outcome measures. Thank you for this helpful suggestion, we have added the following text:

The authors will provide a summary of all the outcomes and associated measurement tools, that have been specifically analysed within the included systematic reviews and meta-analyses. If a review has correlated such study endpoints or outcomes to cancer survival or quality of life; this will also be reported.

---

## [Editor Report · Decision Letter 1]

17 Mar 2025

Protocol for an umbrella review of systematic reviews evaluating the efficacy of digital health solutions in supporting adult cancer survivorship care.

PONE-D-24-26581R1

Dear Dr. Keane,

We’re pleased to inform you that your manuscript has been judged scientifically suitable for publication and will be formally accepted for publication once it meets all outstanding technical requirements.

Kind regards,

Ronald Chow

Academic Editor

PLOS ONE

---

## [Editor Report · Acceptance letter]

PONE-D-24-26581R1

PLOS ONE

Dear Dr. Keane,

I'm pleased to inform you that your manuscript has been deemed suitable for publication in PLOS ONE. Congratulations! Your manuscript is now being handed over to our production team.

Kind regards,

on behalf of

Mr. Ronald Chow

Academic Editor

PLOS ONE